

# Effects of microbial evolution dominate those of experimental host-mediated indirect selection

Jigyasa Arora[1], Margaret A. Mars Brisbin[1] and Alexander S. Mikheyev[1,2]

[1] Okinawa Institute of Science and Technology Graduate University, Onna-son, Okinawa, Japan
[2] Research School of Biology, Australian National University, Acton, ACT, Australia

## ABSTRACT

Microbes ubiquitously inhabit animals and plants, often affecting their host's phenotype. As a result, even in a constant genetic background, the host's phenotype may evolve through indirect selection on the microbiome. 'Microbiome engineering' offers a promising novel approach for attaining desired host traits but has been attempted only a few times. Building on the known role of the microbiome on development in fruit flies, we attempted to evolve earlier-eclosing flies by selecting on microbes in the growth media. We carried out parallel evolution experiments in no- and high-sugar diets by transferring media associated with fast-developing fly lines over the course of four selection cycles. In each cycle, we used sterile eggs from the same inbred population, and assayed mean fly eclosion times. Ultimately, flies eclosed seven to twelve hours earlier, depending on the diet, but microbiome engineering had no effect relative to a random-selection control treatment. 16S rRNA gene sequencing showed that the microbiome did evolve, particularly in the no sugar diet, with an increase in Shannon diversity over time. Thus, while microbiome evolution did affect host eclosion times, these effects were incidental. Instead, any experimentally enforced selection effects were swamped by uncontrolled microbial evolution, likely resulting in its adaptation to the media. These results imply that selection on host phenotypes must be strong enough to overcome other selection pressures simultaneously operating on the microbiome. The independent evolutionary trajectories of the host and the microbiome may limit the extent to which indirect selection on the microbiome can ultimately affect host phenotype. Random-selection lines accounting for independent microbial evolution are essential for experimental microbiome engineering studies.

Corresponding author
Alexander S. Mikheyev,
alexander.mikheyev@oist.jp

## INTRODUCTION

Communities of microbes living on or in multicellular host organisms interact with their hosts in diverse ways that often influence host phenotype and fitness (*Zilber-Rosenberg & Rosenberg, 2008*). Such host-microbe interactions have traditionally been investigated by experimentally comparing hosts raised without a microbiome (axenic) to hosts inoculated with known components of the microbiome (gnotobiotic) or that receive microbiome transplants composed of complex communities (*Turnbaugh et al., 2009*). Observations of

atypical phenotypes in axenic organisms indicate hosts are dependent on their microbiome and cannot function normally without it (*Shin et al., 2011*; *Theis et al., 2016*; *Rosenberg & Zilber-Rosenberg, 2018*). The integral role of the microbiome in shaping host phenotype suggests that desirable host traits can be indirectly selected through microbiome engineering (*Mueller & Sachs, 2015*; *Gopal & Gupta, 2016*; *Oyserman, Medema & Raaijmakers, 2018*). To achieve this, microbes or microbial communities correlated with desired host traits are selected, but selection success is evaluated by measuring host traits (*Mueller et al., 2019*). This novel approach has numerous practical applications, such as better probiotic design and improved crop yields, without requiring selection on genetically diverse hosts or otherwise altering hosts through genetic engineering.

While applying or administering specific bacterial strains or communities (i.e., probiotics) to achieve a desired host effect is now widespread, true microbiome engineering studies remain rare. Diverse examples of successful probiotic studies include: increased biomass and antioxidant capacity in plants inoculated with *Agrobacterium* (*Chihaoui et al., 2015*), reduced white pox disease in corals that received a probiotic cocktail of 13 bacterial strains isolated from coral mucus (*Alagely et al., 2011*), and intestinal epithelial cells with increased ability to keep pathogens from escaping the intestinal tract in mice that were administered *Lactobacillus* strains (*Mack et al., 2003*). While often successful, probiotic approaches typically rely on relatively simple manipulations of the microbiome by introducing known and culturable bacterial species. Additionally, probiotic studies usually take advantage of some prior knowledge of host-microbe interactions involving the host or microbe(s) of interest. However, microbiomes as communities are more complex than what is generally applied experimentally (*Qin et al., 2010*) and can elicit greater magnitude and more specific responses (*Sheth et al., 2016*) than synthetically prepared treatments. In contrast to probiotics, microbiome engineering leverages complex microbial communities by modifying and transferring entire microbiomes, including unknown or unculturable bacterial strains, without prior knowledge of host-microbe interactions by selecting microbiomes based on host phenotype (*Mueller & Sachs, 2015*).

As complex dynamic interactions among microbes in an *in-situ* microbial community are difficult to manipulate, only a few studies have so far tried to engineer native microbiome communities. *Swenson, Wilson & Elias (2000)* first engineered the *Arabidopsis thaliana* rhizosphere microbiome to increase and decrease shoot biomass by inoculating fifteen successive *Arabidopsis* selection rounds with the microbiome of plants with the highest or lowest above-ground biomass in the preceding round (*Swenson, Wilson & Elias, 2000*). *Panke-Buisse et al. (2015)* expanded this application by selecting on late and early flowering-time under nutrient stress and demonstrating that the engineered microbiomes could influence flowering-time in additional *Arabidopsis* strains, as well as another related plant. Importantly, *Panke-Buisse et al. (2015)* evaluated microbiome composition through 16S rRNA amplicon sequencing, clearly illustrating that the microbiome evolved in response to host selection. However, *Mueller & Sachs (2015)* proposed the use of random-selection lines—where the propagated microbiome is randomly chosen from replicates—as the gold standard for experimental controls in microbiome engineering experiments, even while admitting that they greatly increase experimental effort. Previous microbiome

engineering studies relied on sterile media transfers as negative controls, although *Mueller et al. (2019)* also incorporated a fallow-soil control for the presence of naturally occurring microbes. Notably, none of these studies used random-selection controls, which account for independent microbial evolution that may otherwise confound results. Furthermore, microbiome engineering experiments have, to the best of our knowledge, not yet been attempted with animal models.

In this study, we performed a microbiome engineering experiment incorporating random-selection lines the first time in an animal model for. We chose the fruit fly, *Drosophila melanogaster,* as a model for microbiome selection because of its relatively quick generation times (*Trinder et al., 2017*) and its simple core gut microbiome community (<30 major species), which are largely commensals acquired from the environment and transmitted between flies (*Erkosar et al., 2013*; *Blum et al., 2013*). Furthermore, the microbiome has been implicated in a wide range of host-associated functions (*Wong, Ng & Douglas, 2011*; *Ridley et al., 2012*; *Broderick & Lemaitre, 2012*; *Engel & Moran, 2013*); for example, fly development (or eclosion time, *May, Doroszuk & Zwaan, 2015*), immunity, mating, response to external infection, and aging (*Charroux & Royet, 2012*; *Gould et al., 2018*).

Taking advantage of the fact that the microbiome affects fly development time i.e., emergence of adult flies from pupae (*Shin et al., 2011*; *Ridley et al., 2012*), we attempted to select for a microbiome that speeds up fly eclosion in sugar-starved flies and flies fed a high-sugar diet. Over the course of four selection cycles, we propagated the microbiome from vials with fast-eclosing flies and saw a significant decrease in fly eclosion times. However, there was no difference between fly eclosion times in selected treatments and random-selection controls. Rather, the phenotypic changes were a byproduct of microbes adapting to the media instead of the applied selective pressure. Our results emphasize the need for proper controls in microbiome evolution experiments and suggest that independent selection pressures on the microbiome may sometimes dominate in microbiome selection experiments.

## MATERIALS AND METHODS

### Fly maintenance and phenotyping

The *Drosophila melanogaster* strain, Canton S, was used in this experiment because it has been kept inbred since its collection in the early 20th century (*Stern, 1943*) which minimizes potential for host evolution over repeated experimental cycles, e.g., due to drift or selection pressures acting on the stock population (*Emborski & Mikheyev, 2019*). Stock flies were reared on standard media (4% yeast, 8% dextrose, 1% agar, 0.4% propionic acid, 0.3% butyl p-hydroxybenzonate) at 25 °C and 60% relative humidity under 12 hr:12 hr light/dark schedule. The flies have been maintained in the standard diet for 10 years. For the no-sugar diet, sugar and cornmeal were removed, whereas in the high-sugar diet was prepared with an additional 16% sugar. These diets represented different ecological conditions ('famine *vs.* feast'). Fresh media was prepared for each experimental cycle and 25 ml aliquots were distributed to sterile flat bottom vials (23 mm in diameter) for fly

rearing. Three-day old stock flies were mated in egg collection cages with grape juice agar and yeast for 24 h. The agar-media was changed after 24 h (*Koyle et al., 2016*), and fly eggs were collected 8 h after the change in the media, to ensure that eggs were laid within eight hours of each other. The eggs were surface-sterilized by gently rinsing 2x with a solution of double distilled water and 50% bleach for 30–120 s (*Newell & Douglas, 2014*; *Obadia et al., 2018*).

As most fly eclosions from pupae occur during the day, the developmental time of the flies was assessed by recording the number of newly-eclosed flies every hour during the 12 hr light period, from 9:00 AM to 9:00 PM, and discarding them before they could mate with other newly-eclosed flies. Eclosion times were recorded for three days, starting from the eclosion of the first fly. Overall, 10,850 flies were phenotyped in the experiment.

## Indirect selection of the microbiome

We used the experimental protocol suggested by *Mueller & Sachs (2015)* for one-sided artificial selection on microbiomes (Fig. 1). A number of studies have shown interactions between the fly microbiome and fly developmental time and metabolism under varying different nutrient conditions (*Broderick, Buchon & Lemaitre, 2014*; *Ridley et al., 2012*; Jehrke et al., 2018). As a result, we ran the experiment in parallel with two different media types, one without any added sugar and one with fewer carbohydrates (i.e., no cornmeal)". As a result, the experiment was effectively run twice, but in different media. Stock flies that were 24 hr old were collected from stock diet vials (60 females and 40 males in each stock vial) and incubated in fresh no-sugar and high-sugar diet vials (step 1 of Fig. 1) for 3 days to ensure that all flies developed into sexually mature adults and that females had mated The three-day-old adult flies were transferred to fresh treatment media for 24 h to lay eggs and to establish the original microbiome community in high-sugar and no-sugar treatment diets.

Each treatment (selection/high-sugar, selection/no-sugar, no-selection/high-sugar, no-selection/no-sugar) was initiated with 10 replicate lines, each of which was split into three sub-replicates at each selection cycle (30 vials per treatment). For selection treatments, the microbiome from the sub-replicate with the shortest mean eclosion time was selected to inoculate each of the three sub-replicates for that line in the next selection cycle (step 3 of Fig. 1). For no-selection treatments, sub-replicates were randomly chosen for microbiome propagation to the next selection cycle. Microbiome transfer was accomplished by passing the top layer (~1 mm) of the fly food media through a 70 μm-mesh-size cell strainer (Fisher Scientific, cat no. 08-771-19) to remove any dead flies, unfertilized eggs or larvae and then equally distributing the strained media to the food surface in the three sub-replicate vials of the corresponding line in the next selection cycle. The top 1 mm of media was chosen as it is most likely to consist of native fly microbiome from the parent feces (*Wong et al., 2015*). Autoclaved spatulas were used for each food transfer to prevent any cross-contamination between lines. To ensure that the host genotype remained constant and only the microbiome evolved, a spatula of surface-sterilized stock fly eggs was aseptically transferred to each vial using a fresh autoclaved spatula (step 4 of Fig. 1). A spatula of media from vials chosen for

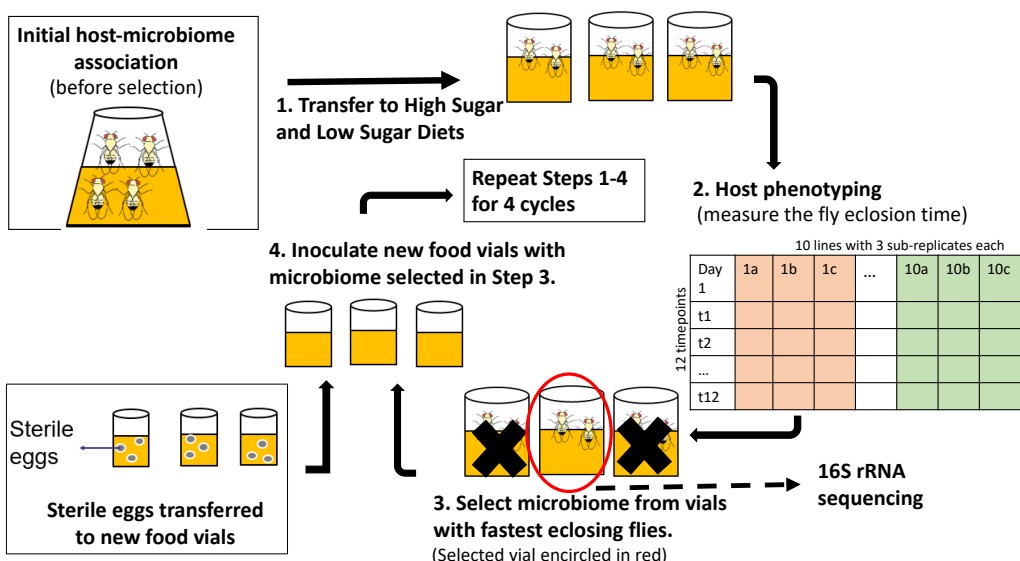

**Figure 1 Schematic of experimental design for indirect selection of trait-associated microbiome in fruit flies.** Following the experimental design suggested by *Mueller & Sachs (2015)*, stock flies laid eggs in either high-sugar or no-sugar diets (step 1) and the microbiome from the fastest eclosing flies was propagated to the next selection cycle (steps 2–4). There were replicate ten selection lines in each treatment, which were split into three sub-replicates during each cycle, which were phenotyped (step 2). To keep host genotype constant, sterile eggs from stock flies were used in each selection cycle (step 4). There were ten parallel lines in each treatment, which were split into three sub-replicates at each selection cycle. Random-selection lines were simultaneously maintained in high- and no-sugar media as experimental controls.

propagation to the next cycle was stored at −80 °C for 16S rRNA gene sequencing. The selection and no-selection procedures were repeated for a total of four selection cycles.

## 16S rRNA gene analysis

DNA was extracted from media collected from the sub-replicate vials chosen for propagation in both selection and no-selection treatments for all rounds and diets. Extractions were performed using the DNeasy Blood and Tissue kit (QIAGEN, Hilden, Germany) following manufacturer's protocols. Library preparation was done using the "16S Metagenomic Sequencing Library Preparation" protocol (Illumina) targeting 16S rRNA gene V3 and V4 regions using Illumina general primer pair. 5% PhiX control was added as an internal control for low diversity libraries. The libraries were sequenced by the Okinawa Institute of Science and Technology (OIST) sequencing section on the Illumina MiSeq platform with 2x250-bp v2 chemistry. The reverse read quality was too poor to join paired-end reads, however, and analysis was carried out on demultiplexed single-end sequences in QIIME 2 (v2017.11, *Bolyen et al., 2018*). The Divisive Amplicon Denoising Algorithm (DADA) was applied through the DADA2 plug-in for QIIME 2 to quality-filter sequences, remove chimeras, and construct the Amplicon Sequence Variant (ASV) feature table (*Callahan et al., 2016a*). We chose to analyze ASVs instead of Operational Taxonomic Units (OTUs) because ASVs are reproducible and reusable across studies, whereas OTUs are study specific (*Callahan, McMurdie & Holmes, 2017*). Taxonomic assignments were

given to ASVs by importing SILVA 16S rRNA representative sequences and consensus taxonomy (release 128, *Quast et al., 2013*) to QIIME 2 and classifying representative ASVs using the naive Bayes classifier plug-in (*Bokulich et al., 2018*). The feature table, taxonomy, and phylogenetic tree were then exported from QIIME 2 to the R statistical environment (*R Core Team, 2013*) and combined into a Phyloseq object (*McMurdie & Holmes, 2013*). To reduce the effects of uncertainty in ASV taxonomic classification, we conducted the analysis at the microbial 'genus' level. Prevalence filtering was applied to remove low-prevalence ASVs with less than 1% prevalence in order to decrease the possibility of data artifacts affecting the analysis (*Callahan et al., 2016b*). Sequence counts were converted to relative abundance to normalize for varied library size and Weighted Unifrac (*Lozupone et al., 2011*) distances were computed between samples. Significance testing for distances between treatment groups was accomplished with the adonis function (Permutational Multivariate Analysis of Variance) in the Vegan R package (*Oksanen et al., 2015*), as well as the DESeq 2 pipeline implemented in phyloseq (*Love, Huber & Anders, 2014*).

## Statistical analysis

Data were analyzed using the R statistical software (version 3.4.0; *R Core Team, 2013*) with tidyr (*Wickham & Henry, 2019*) and ggplot2 packages (*Wickham, 2010*) for data manipulation and visualization. Because fly eclosion was measured hourly, we used the mean eclosion time per vial as a response variable. It was fit as a response in a mixed model against observed effects of diet, cycle and selection (fixed effects) and replicate selection lines within the treatment (random effect) using nlme package (*Pinheiro et al., 2019*). Results were visualized using the Effects package (*Fox et al., 2019*). We visually confirmed distributional assumptions of model fit.

## Data accessibility and analytical reproducibility

All data and code necessary to reproduce the statistical tests, the main figures and tables are available on GitHub (https://github.com/MikheyevLab/drosophila-microbiome-selection), including an interactive online document for the R-based analysis: https://mikheyevlab.github.io/drosophila-microbiome-selection/. Sequence data have been deposited into NCBI SRA database under the accession number PRJNA555001.

## RESULTS

### Fly eclosion time is unaffected by artificial microbiome selection

To examine if diet, selection and cycle leads to faster fly eclosion time, we used linear mixed effects models, which allow for testing nested random effects and within-group variation. We used selection-cycle, diet and artificial microbiome selection as fixed effects, and lines with sample replicates nested within them as random effects. We observed significant contribution of diet and selection-cycle on fly eclosion time, but artificial microbiome selection did not affect the fly phenotype either as a main effect or an interaction (Fig. 2, Table 1). Flies in high-sugar diets took longer to eclose than those in the no-sugar diet, but eclosion time decreased in both diets.

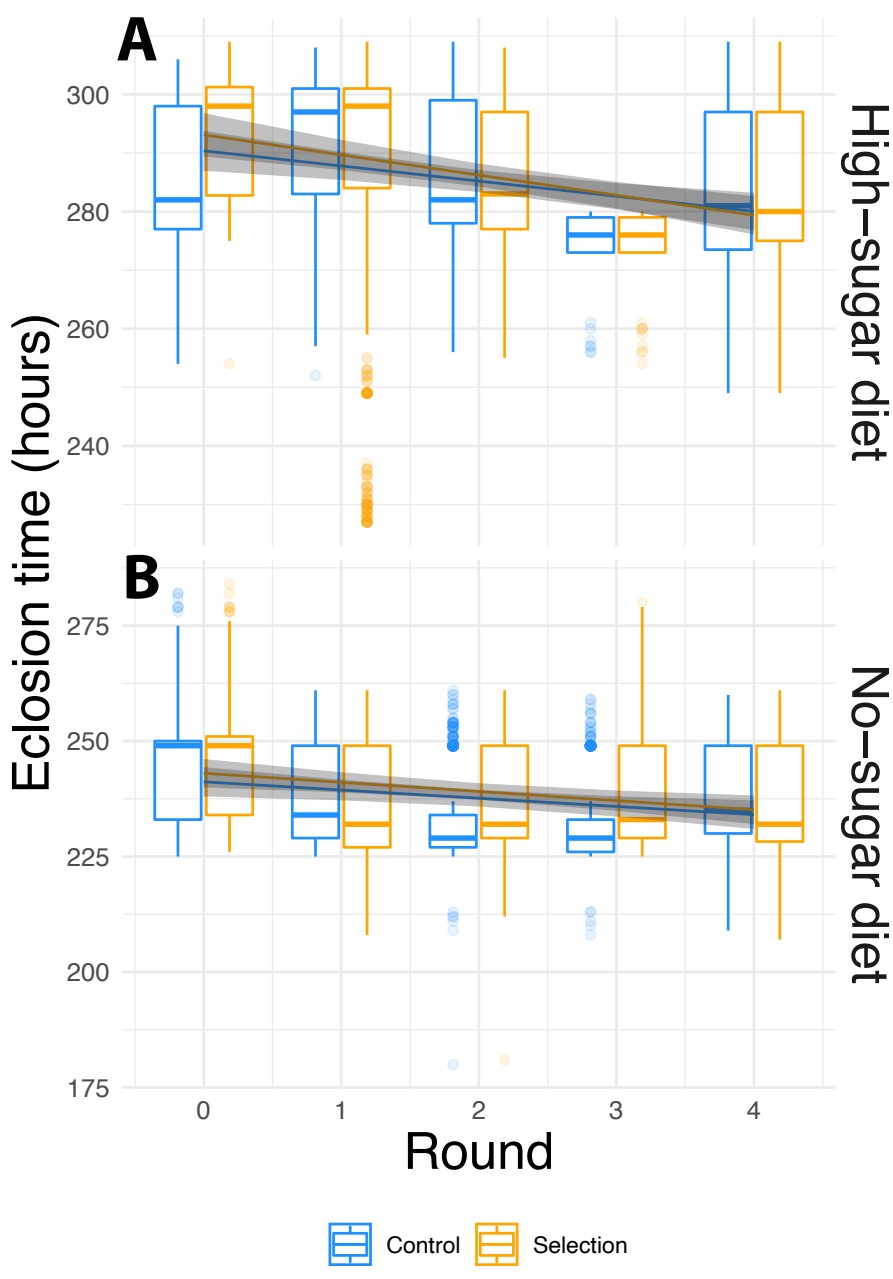

**Figure 2 Phenotypic evolution over the course of the experiment.** (A) High-sugar diet; (B) no sugar-diet. Box plots of raw data, and lines with 95% confidence intervals showing the fit of the mixed-effect linear model for the selection cycle by diet by selection interaction term (Table 1). Box plots hinges show to the first and third quartiles, and whiskers encompass 1.5 times the inter-quartile range. Data beyond the whiskers are are plotted individually. In both diets, fly eclosion times decreased significantly over the course of the experiment. The difference in mean eclosion times between the first and last selection cycle was 7.5 ± 1.2 (S.E.) hours for the no-sugar diet and 12.1 ± 1.2 (S.E.) hours for the high-sugar diet. However, selection had no effect and the rate of decrease was not different for random-selection controls vs. selected flies. Rather than being driven by experimentally enforced selection, changes in phenotype were caused by independent evolution of the microbiome.

**Table 1** While diet and selection cycle had strong effects on eclosion time, the selection did not (also see Fig. 2).

| | Time | | |
|---|---|---|---|
| *Predictors* | *Estimates* | *CI* | *p* |
| Intercept (High-sugar diet, No-selection control) | 292.96 | 288.36–297.57 | **<0.001** |
| Selection cycle | −2.59 | −3.95—1.23 | **<0.001** |
| No-sugar diet | −50.00 | −56.29—43.70 | **<0.001** |
| Selection treatment | 3.58 | −3.21–10.37 | 0.302 |
| Selection cycle × No-sugar diet | 0.81 | −1.06–2.68 | 0.397 |
| Selection cycle × Selection treatment | −0.85 | −2.83–1.14 | 0.402 |
| No-sugar diet × Selection treatment | −1.52 | −10.57–7.53 | 0.742 |
| Selection cycle × No-sugar diet × Selection treatment | 0.65 | −2.03–3.33 | 0.634 |
| $N_{line}$ | 10 | | |
| $N_{replicate}$ | 199 | | |
| Observations | 10,850 | | |
| Marginal $R^2$/Conditional $R^2$ | 0.779/0.834 | | |

## 16S rRNA gene analysis of microbiome composition

We performed 16S rRNA amplicon sequencing of microbiome from both selected and random-selection control media that was chosen for propagation to the next cycle in each diet. We sequenced the V3/V4 hypervariable region of the 16S rRNA gene using the MiSeq v2 platform which generated an average of 175,522 reads per sample. These reads were analyzed using the DADA2 (*Callahan et al., 2016a*) pipeline implemented in QIIME 2 (*Bolyen et al., 2018*). ASVs with low prevalence (<0.01) were removed and alpha-diversity was measured by Shannon-diversity Index that accounts for both species abundance and evenness (*Willis & Martin, 2018*). The association between bacterial alpha-diversity and artificial microbiome selection regime was tested via the adonis function in vegan R package (*Oksanen et al., 2015*), with alpha-diversity as dependent variable and diet, cycle, selection pressure as explanatory variables. The alpha-diversity varied with both diet and cycle. It increased in each successive cycle for both selected and non-selected vials, but it was more pronounced in no-sugar *vs.* high-sugar diet (Fig. 3).

In general, the media contained low bacterial diversity, as reported previously (*Blum et al., 2013*). The microbial communities were homogeneous in the initial rounds of both diets. While *Acinetobacter* and *Staphylococcus* increased in frequency over time in high-sugar diet, change in relative abundance of *Pseudomonas* and *Acinetobacter* led to a significant increase in alpha diversity over time in the no-sugar diet (Table 2, Figs. 3 and 4). An adonis analysis of the UniFrac distances between microbial communities found a significant change over cycles for the no-sugar diet ($F = 6.15$, $p = 0.0004$), but not for the high-sugar diet ($F = 0.62$, $p = 0.73$), consistent with alpha diversity and compositional differences (Figs. 3 and 4). We could not detect specific genera that systematically changed over the course of the experiment in either media using linear models implemented in DESeq2.
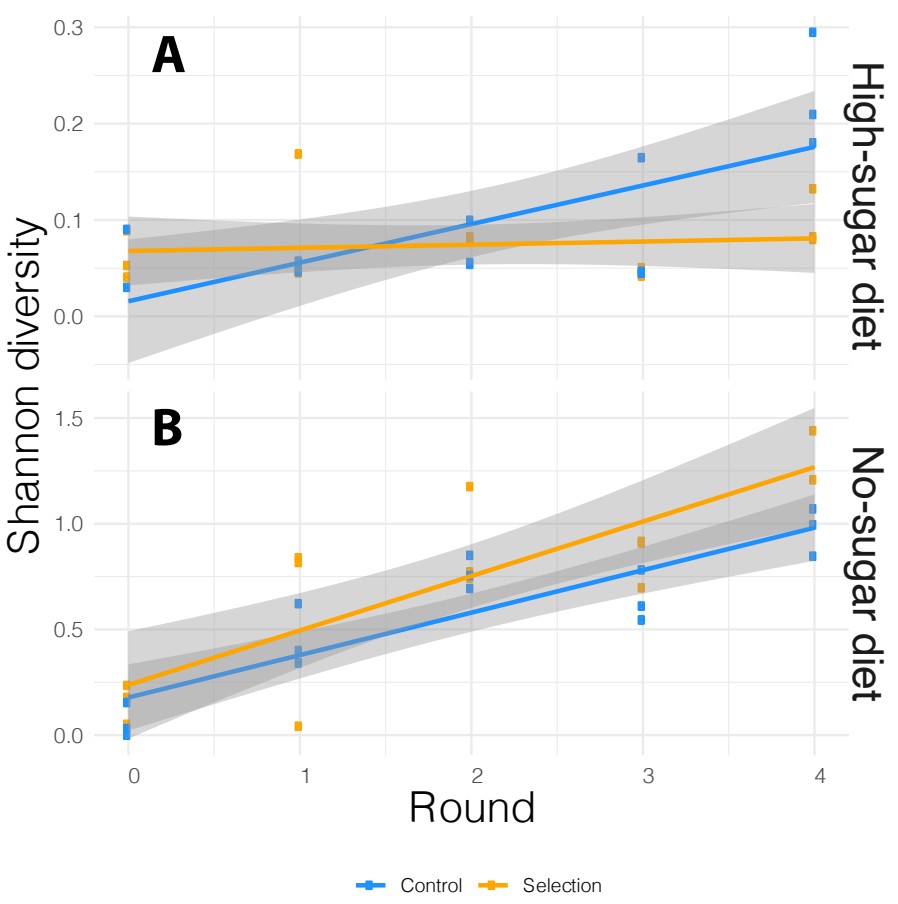

**Figure 3** Alpha-diversity based on Shannon index for the media microbiome in (A) high- and (B) no-sugar diets over four cycles of artificial microbiome selection. Diversity was higher in the no-sugar diet (see also Fig. 4), and increased over time (Table 2). The shaded area represents 95% confidence interval for the line of best fit.

**Table 2** Because there was no effect of selection (Table 1, also see adonis analysis), selection and no-selection treatments were combined to increase power. The alpha diversity was lower in the high-sugar diet (Figs. 3 and 4), and increased over time in the no-sugar diet (Fig. 3).

| Predictors | Value | | |
| --- | --- | --- | --- |
| | Estimates | CI | p |
| Intercept (High-sugar diet) | 0.04 | −0.06–0.15 | 0.420 |
| No-sugar diet | 0.17 | 0.02–0.32 | **0.031** |
| Selection cycle | 0.02 | −0.02–0.06 | 0.353 |
| Selection cycle × No-sugar diet | 0.20 | 0.14–0.26 | **<0.001** |
| Observations | 58 | | |
| $R^2/R^2$ adjusted | 0.833/0.824 | | |

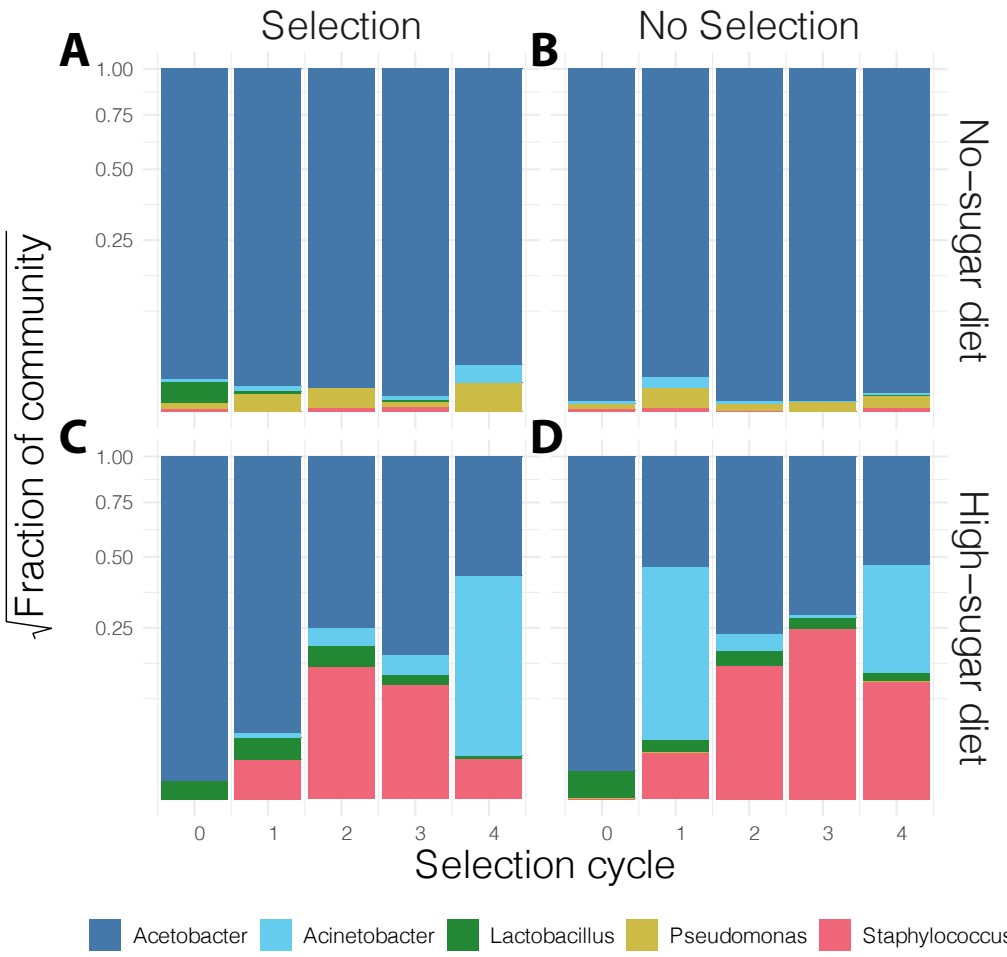

**Figure 4 Stacked bar plot of bacterial genus-level relative abundance in the media over the course of the experiment.** The compositional changes in community structure over time were only significantly different for the no-sugar diet (A, B), compared to the high-sugar diet (C, D) (see also alpha diversity plots in Fig. 3). The data are aggregated across three replicates in each condition and the shaded area represents 95% confidence interval for the line of best fit. This suggests that the microbiome of the two media evolved differently, despite producing similar phenotypic results (Fig. 2).

## DISCUSSION

We attempted to apply microbiome engineering to increase fruit fly development rate by propagating the microbial community associated with fast development over four selection cycles (Fig. 1). We observed substantial increase in developmental rates over the course of the four selection cycles in two different dietary media with no- or high-sugar content. However, selection for more rapid eclosion had no effect on either the developmental rate or its change over the course of the experiment (Fig. 2). 16S rRNA amplicon sequencing showed that the microbial diversity did indeed change over time, with a general increase in alpha diversity, particularly in the no-sugar diet, but it too was unaffected by selection (Figs. 3 and 4). Thus, independent microbial evolution in the media swamped any signal of

experimentally induced selection. This phenomenon is commonly observed in microbes, which rapidly adapt to culture media, but the magnitude and significance of this effect has largely been neglected by microbiome engineering work. However, by chance, the effects of microbial adaptation affected the host phenotype in the same direction as our selection pressure. Only by using a random-selection control could we detect that the entirety of the observed effect was incidental.

To the best of our knowledge, this is the first study to examine host-mediated indirect selection of microbiome in an animal model. The fruit fly is an excellent model for microbiome manipulations. It is an open symbiotic system, meaning that the microbiome is horizontally transferred and is supplemented and maintained through food consumption (*Blum et al., 2013*; *Wong et al., 2015*). As a result, the processes affecting the microbiome can be complex, including a mixture of ecological, evolutionary and social interactions (*Mueller et al., 2005*; *Kaltenpoth et al., 2014*). For instance, behavior of individual flies, such as regurgitation and fecal deposition in the food, tunneling, allo-coprophagy (consumption of conspecifics' feces), and acquisition of fly microbes through food consumption rather than internal maintenance, leads to an exchange of symbionts among group members reared in the same media (*Chandler et al., 2011*; *Goodrich et al., 2014*; *Wong et al., 2015*; *Körner, Diehl & Meunier, 2016*). This aspect of fly biology motivated the media transfer in our experiment. However, since the inoculating flies came from a common stock, the initial microbiome diversity may have been low, with less variation available for subsequent artificial selection. Yet, the microbiome did evolve over subsequent cycles, with significant phenotypic effects on the flies.

It is well-known that the microbiome affects fly nutrition and development, particularly by affecting the amount of fat (triglycerides) in the host; axenic individuals have a longer developmental period (*Storelli et al., 2011*; *Shin et al., 2011*; *Ridley et al., 2012*; *Newell & Douglas, 2014*; *Ma et al., 2015*). However, different bacterial species have unique effects on the host. It is likely that our measures of bacterial relative abundance and community diversity metrics (Figs. 3 and 4) cannot fully capture the complexity of bacterial interactions with the host. This is exemplified by the fact that we only detected significant changes in bacterial community composition in no-sugar media, yet, flies in both media types—high-sugar and no-sugar—had comparable decreases in eclosion times. It is, therefore, possible that phenotypic changes resulted from effects of lower-frequency strains (*Matsutani et al., 2011*; *Chouaia et al., 2014*), or perhaps from other factors, such as chemical compounds produced by bacteria in response to each other or to the media composition. In complex systems, such as microbial communities, substantial phenotypic variation may be due to interactions between its components, which play a role in facilitating community-level selection (*Williams & Lenton, 2007*).

## Implications for the design of experiments using artificial selection of microbiome engineering

### Did the experiment have power to detect effects of artificial selection?

This question may be answered by drawing on a parallel with artificial selection on genes, which is a chief appeal of artificial selection on the microbiome. If there is no

heritable variation in the microbiome's effect on host phenotype, it cannot evolve in response to selection. In the context of classical genetics, to understand whether evolution is possible in principle, key parameters include (1) the selection differential ($S$), which is the difference in average trait values between selected and unselected individuals and (2) the heritability ($h^2$). Given these values, the short-term response is governed by the breeder's equation R = $h^2 S$ (*Falconer & Mackay, 1996*). In our experiment correlation between the 'parental' and 'offspring' mean phenotypes measured in the absence of selection was high ($r = 0.92$), indicative of a high 'microbial heritability' ($h^2_m$) in the system. Likewise, by selecting the top third of the replicates we attained a strong selection coefficient (6.8 h $\pm$ S.D. 11.5/selection cycle). Based on these considerations alone, we would expect to see measurable changes at the end of the experiment in the selected treatments. Under ideal circumstances mean eclosion time could potentially have been reduced by over a day. However, in our treatments eclosion was accelerated by about half a day. Note, that the parameter $h^2_m$ is only broadly analagous to narro-sense genetic heritability, since the microbiome is potentially more mutable, so these calculations are only crude approximations. Nonetheless, estimating the amount of expected change in response to selection is key before embarking on a long-running experimental selection design, and we strongly recommend a pilot study to estimate $h^2_m$ and $S$ beforehand. If there is no $h^2_m$ in the system, a selection experiment will not likely succeed.

## Efficacy of experimental selection vs. independent evolution by the microbiome: implication for controls

Microbial evolution experiments typically apply discrete cycles of selection (*Swenson, Wilson & Elias, 2000*; *Panke-Buisse et al., 2015*; *Mueller et al., 2019*). However, media microbiome evolved continuously between selection cycles and not necessarily in ways that we wanted or could effectively control. For example, to be passed to the next selection cycle, microbes had to aggressively colonize fresh media and compete amongst themselves for resources, but not in a way that negatively affected fly larvae. Because there are many microbial generations within selection cycles, these parallel selection pressures may dominate the evolutionary response with significant effects on the host phenotype, as appears to have happened in our experiment.

We did not anticipate the strength of independent evolution by the microbiome when designing our study, and the topic has received relatively little theoretical or empirical attention (though see *Williams & Lenton, 2007*). One key implication is for the design of controls during microbiome evolution studies. Randomly selected control lines allow the microbiome to evolve in the same way, except for the experimentally enforced selection. However, these controls are extremely time- and labor-consuming. Alternative options, such as constant inoculation from a preserved microbiome source (*Martino et al., 2018*) or null inoculations, have been proposed as efficient alternatives (*Mueller & Sachs, 2015*). Even the fallow-soil control used by *Mueller et al. (2019)*, which is a substantial methodological advance over typically used sterile controls, does not take into account possible interactions between the microbiome and the plant and how they might evolve. Experimental designs with other control strategies do not provide the same level of control over microbial

evolution as does the random-selection control. For example, using constant or null controls in our experiment would have led us to erroneously infer that the microbiome evolved in response to experimental selection. Understanding the evolutionary processes that can take place during micribiome harvesting, transfer and selection is key to for optimizing microbiome engineering experiment design.

Evolution can take place either by evolution of the initial communities, by changes in the frequencies of existing microbes or by immigration of new ones. Microbiome engineering strives to harness the first two mechanisms, which can be quantified via sequencing protein coding genes along with 16S rRNA (*Matsutani et al., 2011*; *Chouaia et al., 2014*). On the other hand, microbial immigration (i.e., contamination) can be limited, but close to impossible to eliminate unless the experiment takes place in completely sealed systems, such as bioreactors. In more realistic settings where microbial evolution takes place alongside a living host, eliminating immigration is much harder. Specifically, immigration may provide an inherent problem in open systems like the fruit fly gut/media or plant/soil systems, where experimental selection will have to overcome its effects. By contrast, microbiome engineering may be more powerful in closed systems with strict vertical transmission.

## Control of host genotypes

Along similar lines, we cannot exclude the possibility that the host has changed in the course of the experiment. Strictly controlling the host population (e.g., in a glycerol stock or seed bank) is not possible with fruit flies. In retrospect, it would have been desirable to confirm stability of eclosion times in the source population at the beginning and the end of the experiment. Changes in the host population appear a less likely explanation for the observed data, given the magnitude of change seen in the experiment—about half a day earlier eclosion in the course of four selection cycles (Fig. 2). First, the fly stocks were inbred and genetically homogeneous, minimizing the possibility of evolutionary changes (*Emborski & Mikheyev, 2019*). Second, they were kept in a stable environment with controlled temperature, humidity, photoperiod and diet. Third, eggs were surface sterilized to prevent the introduction of additional microbes to the experiment. Nonetheless, even because the most stable-seeming environments, such as glycerol stock or seed banks may experience change over time (e.g., due to freezer malfunctions or fungal rot), ideally both host and microbiome changes should be controlled in the course of microbiome engineering experiments. Therefore, we strongly recommend that studies introduce this 'host-stability' control.

## CONCLUSIONS

In conclusion, the findings show that artificial microbiome selection is not significantly correlated with fly phenotype or microbiome. This was made possible due to the use of random-selection controls to measure selection pressure. The lack of significant correlation of selection might be driven by factors independent of host-mediated artificial selection. Any future prospects in artificial engineering of host microbiome to select desirable host

phenotype would require selection regimes that are stronger than microbial evolution. In short, we recommend the following three considerations for experimental design:

1. **Random-selection controls are mandatory**, as they control for microbiome evolution, either outside the control of the host or through immigration.
2. **A control for the stability host phenotype over the timescale of the experiment** assures that the evolutionary dynamics are, in fact, due to microbiological changes
3. **A power analysis before the start of the experiment** to assure transmission of the phenotype via the microbiome (a measure of heritability) and to compute the extent of phenotypic variability within a selection round. These will help choose an appropriate selection coefficient to obtain a detectable magnitude of response.

## ACKNOWLEDGEMENTS

We are grateful to Carmen Emborski and Ulrich Mueller for help in designing this study. We thank Takakazu Yokokura and Cecelia Lu for providing invaluable training and assistance in fly-rearing and egg collection. We thank the OIST DNA sequencing section—Onna, Okinawa, for carrying out the sequencing.

### Funding

This work has been funded by the Okinawa Institute of Science and Technology Graduate University. The funders had no role in study design, data collection and analysis, decision to publish, or preparation of the manuscript.

### Grant Disclosures

The following grant information was disclosed by the authors:
Okinawa Institute of Science and Technology Graduate University.

### Competing Interests

Alexander S Mikheyev is an Academic Editor for PeerJ.

### Author Contributions

- Jigyasa Arora and Margaret A. Mars Brisbin conceived and designed the experiments, performed the experiments, analyzed the data, prepared figures and/or tables, authored or reviewed drafts of the paper, and approved the final draft.
- Alexander S. Mikheyev conceived and designed the experiments, analyzed the data, prepared figures and/or tables, authored or reviewed drafts of the paper, and approved the final draft.

### DNA Deposition

The following information was supplied regarding the deposition of DNA sequences:
Sequences are available at NCBI: PRJNA555001.

## Data Availability

Data is available at GitHub: https://github.com/MikheyevLab/drosophila-microbiome-selection.

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
