# Peer review of "Effects of microbial evolution dominate those of experimental host-mediated indirect selection"

_PeerJ, doi:10.7717/peerj.9350_

## Round 0.1 · original submission · Major Revisions

While all three reviewers were generally complimentary of the experimental design and conceptualization and execution of the study, there was strong criticism of key aspects that certainly preclude publication of the manuscript in its current form and may preclude publication even after revision. In particular, the transfer method of the microbiome was a mix of both fecal communities and those growing in the media, making it difficult to know if the microbiome used was biologically relevant. Reviewer 3 feels strongly that this is an insurmountable experimental flaw, and the authors should address this point explicitly, thoroughly and clearly, making certain to address it in the manuscript directly and making a strong case why this issue does not invalidate the work if they wish to see it published in PeerJ. Reviewer 2 also felt that given the types and dynamics of microbiomes found there was little evidence to support the claims in the title and abstract. The other two reviewers raised additional significant issues, including a tendency to overextrapolate the findings in the title and abstract (particularly, as R2 reiterates, because the changes in the microbiome cannot clearly be attributed to the experimental treatments, or more work should be done to clarify if they can) and much more detail on the composition of the media (which pertains directly to the point of R3).

Reviewer 1 ·

Basic reporting

- Literature used is relevant. However, the authors fail to cite or discuss the following paper, which directly addresses similar questions in the same model organism:
o Bacterial Adaptation to the Host’s Diet Is a Key Evolutionary Force Shaping Drosophila-Lactobacillus Symbiosis Martino et al., Cell Host & Microbe 2018 https://doi.org/10.1016/j.chom.2018.06.001
This is a major oversight and should be addressed.
- The title of the article sounds catchy but implies a conclusion that is not supported by the experimental results. The authors showed that the microbiota composition changed over the course of the experiment, but they did not determine whether it evolved in response to their experimental manipulation or simply changed due to random chance. Even if the microbiota changed in response to selection (be it the “random” control or other), the study did not assess how they changed (e.g. faster growth rates in the diet) or what aspect of the experimental regime they responded to. Simply put, the manuscript does not answer the question “what does the microbiota want?” Neither does it demonstrate that microbial evolution ‘overtook’ the effects of indirect selection. The data show that indirect selection and microbial evolution were not correlated, but without more info about what exactly changed in the microbiota it is too speculative to say those changes ‘overtook’ or overpowered the impacts of indirect selection.

Experimental design

- Description of the methods: A few important details are missing from the methods.
o Was the humidity controlled? If so, at what value? Humidity can have significant impacts on developmental rates in Drosophila.
o The recipe for the fly diet used is not complete. At present it states “..reared on standard media with 45g of cornmeal/100g of sugar.” Then says “For no-sugar diet, sugar and cornmeal were removed.” This would suggest the no-sugar diet contained nothing! A full accounting of the ingredients in the diet is needed for the reader to evaluate the manuscript. In particular – did either the stock vials or experimental ones contain preservatives? Also, did they contain yeast? Many labs use live yeast in their fly cultures, which is a dimension of the microbiome that you did not assess.
o About how many flies developed per vial? This is an important figure to include because of the effects that density can have on development time.
o Were samples pooled from the three sub-replicate vials for 16S sequencing? This seems to be the case, but it is not specifically stated.
- 16S sequences do a very poor job distinguishing between Acetobacter species (and even between related genera in the Acetobacteraceae) See references below. So, it is quite possible that the microbiome changed a lot on the no-sugar diet but you did not see it based on the resolution of the tools used.
o Matsutani, M., Hirakawa, H., Yakushi, T., & Matsushita, K. (2011). Gen- ome-wide phylogenetic analysis of Gluconobacter, Acetobacter, and Gluconacetobacter. FEMS Microbiology Letters, 315, 122–128.
o Chouaia, B., Gaiarsa, S., Crotti, E., Comandatore, F., Degli Esposti, M., Ricci, I., ... Daffonchio, D. (2014). Acetic acid bacteria genomes reveal functional traits for adaptation to life in insect guts. Genome Biology and Evolution, 6, 912–920.

Validity of the findings

- Rigor of the investigation.
o The random selection control is key and the main conclusion of the study hinges on the fact that it changed in the same direction and magnitude as the experimental population. However, it is hard to assess the significance of the change observed without additional replication (see below) or an assessment of the variance in the development time of stock flies or of axenic animals over the same timeframe (what the authors refer to in the discussion as a ‘host-stability’ control). Having some independent development data would clarify the extent to which the changes observed were the result of the experimental manipulations vs. chance environmental factors.
o The lack of replication is a serious concern for me. If there were parallel technical replicates whose microbiota were passaged and sequenced independently, it would be possible to see if the changes that occurred in the microbiota were non-random (i.e. reproducible and resulting from selection – via diet or host). Also, if there had been independent replication (same experiment at a different time), this would enable us to see if the selection regime (random or not) had the same biological effects on distinct groups of flies and microbes. One example of where significant variance can arise in experiments tracking development time of gnotobiotic Drosophila is the circadian rhythmicity of oviposition. The time of day during which egg laying is initiated and the time when collection occurs prior to surface sterilization influences the age of embryos at the start of the experiment. Given that a 24 hour window was used there could be enough variation in this window to produce the differences you observed in eclosion time across selection rounds. Was the same time of day used for these steps during each round?

Additional comments

Line 62 – you seem to define microbiome engineering using the word engineering. Would ‘modifying and transferring…” or “evolving and transferring…” work here?

Line 93 – Whether Drosophila melanogaster has a core gut microbiome is a matter of debate.
Wong et al. essentially show that the microbiome is inconstant and there is not a core microbiome in lab flies. (Wong AC, Chaston JM, Douglas AE. ISME J. 2013 Oct;7(10):1922-32. doi: 10.1038/ismej.2013.86) This view has critical implications for your experimental design because the starting microbiome likely represented a random subset of the strains in the laboratory population.

Line 261 – This seem an apt place to discuss Martino et al. (mentioned above). Your manuscript does not mention an estimated number of bacterial generations that passed (though you speculate “many” at line 328). This is relevant if significant adaptation to the culture medium was occurring that the level of individual species vs. changes in the relative abundance of species in the microbiome, or both.

Lin 266 – Host-mediated (hyphen)

Line 284 – “interacting” is an odd word choice here

Line 281 (and elsewhere) Your language causally links the changes in the microbiome to changes in fly phenotypes. However, your data do not show this causality. They show a correlation. Without replication (see comments above) or the ‘host stability’ controls referenced in the discussion, there is still doubt that the microbiome change caused the change in fly phenotype.

Figure 1: There are two “step 2” and no “step 3” in the diagram

Figure 2: The labels in the key are confusing. Either get rid of the first “selection” in the diagram, or keep it and change the other labels to “random” (instead of control) and “phenotypic” or something along those lines.

Reviewer 2 ·

Basic reporting

The manuscript by Arora et al. reports findings from a microbial engineering experiment wherein experimental evolution is used to select for microbes that reduce Drosophila melanogaster eclosion time. The results show that it was possible to select for faster eclosing Drosophila lines; however, the microbiome was not a contributing factor to host developmental acceleration despite a change in microbiome composition over the selection cycles.

A major strength of the manuscript is that a random-selection control rather than a null inoculation control was used. Overall, the manuscript is well-written and the results support the conclusions. The raw data and code to produce the figures and statistical findings have been shared.

Please add the following references to Line 97:
Wong, C. N. A., Ng, P. & Douglas, A. E. Low-diversity bacterial community in the gut of the fruitfly Drosophila melanogaster. Environ. Microbiol. 13, 1889–1900 (2011)
Ridley, E.V. et al., 2012. Impact of the resident microbiota on the nutritional phenotype of Drosophila melanogaster. PloS One, 7(5), p.e36765.

Please clarify the following issues:
Figure 2: I am guessing that each of the circles appearing below or above the plots are the individual replicates. If this is the case, why do only some of the box plots have circles? The circles should be described in the legend.
Figure 3: According to the methods, 3 sub-replicates are made from each selection cycle. Does each circle represent a sub-replicate? If so, why is it the case that only 2 circles appear for some rounds?

Minor corrections:
223-230: Many of these details have already been described in the Methods section. Please edit to remove redundancy.
Figure 1: Step 2 appears twice; there is no Step 3. Also, the table for “Step 2” is very faint and may not reproduce well.

Experimental design

The research question is well-defined and of interest to the field. The experimental design is sound.
The following should be addressed in the methods:
118: It would be helpful if the authors provided a brief rationale for why high-sugar and no-sugar diets were used.
137: Details about the media and numbers of flies placed in each vial should be provided since these parameters can affect host life history traits and microbiome composition. In particular, the authors should address the following:
How many flies were placed in each vial and was this number kept consistent across all conditions and selection cycles?
Was the ratio of males: females constant across all conditions and selection cycles?
Please provide recipe for fly media in terms of % (w/v) rather than “g”.
What was the volume of food placed in each vial and what were the dimensions of the vial?
What was the temperature and humidity of the incubator?

Validity of the findings

Shortcomings and alternate explanations are provided in a thoughtful discussion section. Many in the field will appreciate the conclusions section, especially with respect to recommendations for experimental design.

Reviewer 3 ·

Basic reporting

Arora and colleagues – PeerJ Review –

“The microbiome wants what it wants: microbial evolution overtakes experimental host-mediated indirect selection. “


The manuscript, entitled ‘The microbiome wants what it wants: microbial evolution overtakes experimental host-mediated indirect selection’, by Arora, Brisbin, and Mikkeyev, examined the effects of ‘Microbiome Engineering’ on the microbial flora of fruitflies in an experimental setting.

The paper describes a method in which host phenotypes are experimentally selected upon in fruitflies (developmental time) using clonal population of hosts. Hosts are genetically identical, so the only phenotypic variation that can be selected upon is driven by community differences in the hosts’ microbial flora. The experiment selected on host developmental time in two diet backgrounds, used replication, and employed control lines in which no selection occurred. Microbes were transferred from the feeding dishes / media where the flies lived so included a mix of fecal-generated communities and other flora that were growing in the media. In my opinion this transfer method was the fundamental flaw in the experiment such that the experiment did not produce any of the expected results (or really any results that made sense given the protocol). Because of this flaw little / none of the empirical data are of publishable quality. I found this really disappointing, as the experiments were otherwise fascinating.

The authors take this challenge somewhat in stride and then use the paper to motivate how to better design microbiome-engineering experiments. I found this discussion thoughtful and quite coherent, but it unfortunately gave short shrift to the central flaw in the experiment: that the community that was passaged was incorrectly designed to passage the gut flora community of the flies. Rather than passing the community of microbes that was in the gastrointestinal (GI) tract of the flies, they passaged the community that was on the feces / food /media which likely included a different mix of genotypes than would be optimal in the GI tract. This was clarified within the paper but not at all emphasized in the key points (it came near the end – see below). I really struggle with whether -- and in what venue -- data like this are useful for a journal like PeerJ.

Detailed comments.
1. The abstract and throughout the paper the authors describe the term ‘independent microbial evolution’. This needs to be better defined. It might be better described as community drift except for the fact that the phenotype of the hosts did move in one consistent direction. See esp. Abstract / lines 85-86 / line 257.
2. Lines 36-37. I do not really agree with this statement. It seems like a matter of opinion though, especially since no reference or specific data are offered.
3. Lines 103-108. Given that the authors are explaining the results at this juncture, some explanation should be given of how this shift occurred even though there was no treatment effect.
4. Line 115 The authors should explain what they mean by minimizing host evolution, perhaps my saying something about segregating variation.
5. Lines 135-136. What motivates the diet treatments?
6. Lines 154-155. I cannot understand why this method was used. Given the extreme amount of effort that shaped the rest of the experiment, this seemed like such an unexpected choice rather than grinding up the guts of a few flies to passage. Is there something that I am missing?
7. Lines 182- 183. The benefit of using CSVs is to gain extra resolution over binning genotypes in OTUs. This method is circular in generating CSVs but then binning them at the genus level, which seems very strange. What motivates this?
8. Lines 195-197. Data exploration sounds worrisome in a big -data setting like this. I trust these authors fully, but It would be better to explain exactly what was done and the motivation for each analysis. The statistical analysis should not be shifted to a website that is not part of the publication.
9. Large parts of the results should be moved to the materials and methods section including Lines 212-215, 223-233
10. Lines 304-316 seem unnecessary
11. Lines 325-331 is when the authors finally spell out the fundamental flaw in the experiment. This should be moved to the top and then the rest of the paper will make a lot more sense.

Experimental design

As explained above, the core passaging protocol was flawed.

Validity of the findings

Not sure if this is applicable, since there no findings that linked to the protocol used.

---

## Round 0.2 · Minor Revisions

While the revision reviews took a great deal of time, we finally found a highly qualified senior reviewer to evaluate all of the previous reviews and the revised manuscript. Please address all of the recommendations of this new reviewer in a revised version, paying particular attention to incorporating some of these ideas into your manuscript particularly to address the original, strong, critiques of original Reviewer 3, which the new reviewer has made many helpful comments on reframing and clarification.

Reviewer 2 ·

Basic reporting

No comment.

Experimental design

No comment.

Validity of the findings

No comment.

Additional comments

The authors have done a thorough job of addressing my concerns.

Reviewer 4 ·

Basic reporting

Arora PeerJ43924_UGMReview20Apr2020

I evaluate here the comments of the 3 previous reviewers, the authors' rebuttal letter, and the revised manuscript. I have only minor comments on the main manuscript, all of which are easy to address, and my comments below address mainly the reviewer comments.

The authors' rebuttal letter addresses all reviewer comments in great detail. I disagree with some reviewer comments, but I also agree with a few reviewer comments. The reviewer comments are very constructive and overall positive, and I agree that this manuscript is appropriate for publication in PeerJ.

Agree with reviewer comments:
I agree with Reviewer 1 that the title does not convey quite the main finding, particularly the part "The microbiome does what it wants", and I suggest to shorten the title to simply "Effects of microbial evolution dominate those of experimental host-mediated indirect selection". The microbiome "does what it wants" if allowed to roam on a long leash (per analogy with Foster et al 2017 Nature), but microbiomes on a short leash (i.e., microbiomes under greater host control) may not be able to change uncontrolled because short-leashed microbiomes are shaped significantly by host control, and host phenotype is a more reliable indicator of microbiome properties. Specifically for the study here, if the authors had chosen a slightly different microbiome harvesting scheme (e.g., harvesting microbiomes directly from guts by macerating flies; or by transferring flies to fresh sterile media at the end of a cycle, then allowing the flies to defecate for a very short time, then immediately harvesting the defecated microbiomes before the microbiomes can restructure/evolve outside the gut), the authors may have gotten a greater response to microbiome selection. One key message of the entire study is that the microbiome harvesting and microbiome transfer steps are very important for microbiome selection to work, and these steps will be improved by those who will build on the present study.
Anyway, changing the title is easy to address.

Comments by all 3 reviewers are very constructive, and the authors have addressed all comments in the revision.

Disagree with reviewer comments:
I disagree with Reviewer 3 that the "transfer method was a fundamental flaw" and that therefore "none of the data are of publishable quality". I agree with Reviewer 3 that the microbiome harvesting/transfer methods can be improved, now that we know post-hoc that the particular transfer method chosen by the authors was not optimal (the methods could have been adequate, noone knew because noone tried such an experiment so far), but the study is definitely worth publishing because the study develops and test an experimental protocol that will be used widely by other researchers aiming to engineer gut-microbiomes through differential microbiome propagation in insects and other animals.
If the hologenome concept has any value, then it must be true that we should be able to select on host-microbiome associations (or on microbiomes if host genotype is held constant) and thereby obtain a response to selection in host phenotype, mediated through changes in microbiomes. The authors develop here a protocol to do that for an animal, pinpoint an important step in the protocol (microbiome harvesting/transfer), and the findings will therefore enable others to improve the protocol. This advances the field significantly in this fast-moving and economically important field of microbiome engineering.

My evaluation of the revised manuscript:
The manuscript is a significant and timely addition to the field; as far as I know, it is indeed the first test of host-mediated microbiome selection in an animal.
The writing is very clear and informative.
Some very important insights emerge in the discussion, for example the authors' framing of host-mediated selection within a quantitative-genetic framework (the section "Did the experiment have power to detect effects of artificial selection?"). One challenge of microbiome selection is a proper conceptualization of microbiome heritability, which is unlikely completely analogous to heritability of genomes (as in the breeder equation discussed by the authors). To avoid irking quantitative geneticists here in the discussion, it may be worth pointing out in half a sentence that the narrow-sense heritability in the breeder equation is different from the heritability of a microbiome (or maybe define h with a subheading m for microbiome, to indicate that the heritability here is something different). The heritability of a microbiome is less similar to the heritability of a genome, and more similar to the heritability of an infectious virus with a highly fragmented genome that gets reshuffled/recombined as multiple viruses infect the same cell, and fragments are added or lost with each replication cycle. Heritability of such viruses is difficult to conceptualize, but such viruses clearly evolve, as do microbiomes under host-mediated differential microbiome propagation.
Line 468: A good point, microbiome evolution in the absence of host control will reduce the efficacy of differential microbiome propagation, and therefore it will be important to transfer microbiomes efficiently between hosts. I disagree, however, that the possibility of microbiome evolution in the absence of host control "limits the utility of microbiome engineering"; instead, the main message here should be that, to engineer microbiomes through differential microbiome propagation, it will be important to devise experimental tricks or propagation schemes that minimize microbiome evolution in the absence of host control (that is, it will be important to devise propagation schemes that keep the microbiome always and consistently on a short leash). Maybe the first sentence should be rephrased to highlight less any limits if a protocol is not optimized, but highlight the importance of optimizing all steps in a selection cycle, including microbiome harvesting and transfer.

I was unable to view Figure 1, which was not included in the word.doc sent me for review.

Minor comments; all of these will be easy for the authors to fix:
Line 18: "... but microbiome engineering had no effect relative to a control treatment" OR: "... but microbiome engineering had no effect relative to a random-selection control treatment" OR the phrasing used in lines 128&129.
Line 32: Zilber-Rosenberg is misspelled several times in the manuscript (including the list of references) without the caps in Zilber-rosenberg
Lines 49, 104, 445, 476: The Mueller et al bioRxiv manuscript was updated in 2019, and the list of authors changed: Mueller, U.G., T.E. Juenger, M.R. Kardish, A.L. Carlson, K. Burns, C.C. Smith, D.L. Des Marais. Artificial microbiome-selection to engineer microbiomes that confer salt-tolerance to plants. bioRxiv 081521. doi: biorxiv.org/content/10.1101/081521v2
Lines 339&340: italicize bacterial genera
Line 379: delete redundant parenthesis "("
Line 477: ... does not ....
Line 483: there are three interacting processes, not only two: evolution of microbial genomes of each microbial type/species; changes in the relative abundances of microbial types/species; immigration/extinction of microbial types/species. Immigration/extinction can be viewed as extremes of change in relative abundance (zero -> above zero; above zero -> zero), but I agree it is helpful for an experimenter to separate out immigration/extinction.
Line 484: this is an incomplete sentence. The entire paragraph needs some revision. Microbial immigration can be eliminated in closed systems/mesocosms, even with Drosophila; working with closed systems is experimentally more cumbersome (and less realistic), but not impossible.
Line 568: Random selection lines are important if there is only one selection treatment, as in the present experiment. An alternate experimental approach is two different selection treatments run in parallel in the same experiment, and these two selection treatments effectively serve as controls for each other, then cross selected microbiomes and treatments at the end of the experiment after several rounds of selection (as in the Mueller bioRxiv experiment); for such selection scheme, random selection lines would not be necessary (though of course could also be added if resources/time permit).

Experimental design

Overall sound. See my comments above how a future study can be improved by optimizing the microbiome harvesting and microbiome transfer steps when selecting on gut microbiomes of Drosophila.

Validity of the findings

Valid conclusions, given the data

Additional comments

Some very important insights emerge in the discussion, for example the authors' framing of host-mediated selection within a quantitative-genetic framework (the section "Did the experiment have power to detect effects of artificial selection?"). One challenge of microbiome selection is a proper conceptualization of microbiome heritability, which is unlikely completely analogous to heritability of genomes (as in the breeder equation discussed by the authors). To avoid irking quantitative geneticists here in the discussion, it may be worth pointing out in half a sentence that the narrow-sense heritability in the breeder equation is different from the heritability of a microbiome (or maybe define h with a subheading m for microbiome, to indicate that the heritability here is something different). The heritability of a microbiome is less similar to the heritability of a genome, and more similar to the heritability of an infectious virus with a highly fragmented genome that gets reshuffled/recombined as multiple viruses infect the same cell, and fragments are added or lost with each replication cycle. Heritability of such viruses is difficult to conceptualize, but such viruses clearly evolve, as do microbiomes under host-mediated differential microbiome propagation.
Line 468: A good point, microbiome evolution in the absence of host control will reduce the efficacy of differential microbiome propagation, and therefore it will be important to transfer microbiomes efficiently between hosts. I disagree, however, that the possibility of microbiome evolution in the absence of host control "limits the utility of microbiome engineering"; instead, the main message here should be that, to engineer microbiomes through differential microbiome propagation, it will be important to devise experimental tricks or propagation schemes that minimize microbiome evolution in the absence of host control (that is, it will be important to devise propagation schemes that keep the microbiome always and consistently on a short leash). Maybe the first sentence should be rephrased to highlight less any limits if a protocol is not optimized, but highlight the importance of optimizing all steps in a selection cycle, including microbiome harvesting and transfer.

I was unable to view Figure 1, which was not included in the word.doc sent me for review.

Minor comments; all of these will be easy for the authors to fix:
Line 18: "... but microbiome engineering had no effect relative to a control treatment" OR: "... but microbiome engineering had no effect relative to a random-selection control treatment" OR the phrasing used in lines 128&129.
Line 32: Zilber-Rosenberg is misspelled several times in the manuscript (including the list of references) without the caps in Zilber-rosenberg
Lines 49, 104, 445, 476: The Mueller et al bioRxiv manuscript was updated in 2019, and the list of authors changed: Mueller, U.G., T.E. Juenger, M.R. Kardish, A.L. Carlson, K. Burns, C.C. Smith, D.L. Des Marais. Artificial microbiome-selection to engineer microbiomes that confer salt-tolerance to plants. bioRxiv 081521. doi: biorxiv.org/content/10.1101/081521v2
Lines 339&340: italicize bacterial genera
Line 379: delete redundant parenthesis "("
Line 477: ... does not ....
Line 483: there are three interacting processes, not only two: evolution of microbial genomes of each microbial type/species; changes in the relative abundances of microbial types/species; immigration/extinction of microbial types/species. Immigration/extinction can be viewed as extremes of change in relative abundance (zero -> above zero; above zero -> zero), but I agree it is helpful for an experimenter to separate out immigration/extinction.
Line 484: this is an incomplete sentence. The entire paragraph needs some revision. Microbial immigration can be eliminated in closed systems/mesocosms, even with Drosophila; working with closed systems is experimentally more cumbersome (and less realistic), but not impossible.
Line 568: Random selection lines are important if there is only one selection treatment, as in the present experiment. An alternate experimental approach is two different selection treatments run in parallel in the same experiment, and these two selection treatments effectively serve as controls for each other, then cross selected microbiomes and treatments at the end of the experiment after several rounds of selection (as in the Mueller bioRxiv experiment); for such selection scheme, random selection lines would not be necessary (though of course could also be added if resources/time permit).

---

## Round 0.3 · accepted · Accept

We are pleased to accept this after a very rigorous and thoughtful review process. Thank you for your submission.